# Assessing Walking Programs in Fibromyalgia: A Concordance Study between Measures

**DOI:** 10.3390/ijerph19052995

**Published:** 2022-03-04

**Authors:** Sofía López-Roig, Carmen Ecija, Cecilia Peñacoba, Sofía Ivorra, Ainara Nardi-Rodríguez, Oscar Lecuona, María Angeles Pastor-Mira

**Affiliations:** 1Department of Behavioral Sciences and Health, University Miguel Hernández, 03540 San Juan de Alicante, Spain; slroig@umh.es (S.L.-R.); anardi@umh.es (A.N.-R.); mapastor@umh.es (M.A.P.-M.); 2Department of Psychology, Rey Juan Carlos University, 28922 Madrid, Spain; cecilia.penacoba@urjc.es (C.P.); oscar.lecuona@urjc.es (O.L.); 3Official College of Nursing, 03007 Alicante, Spain; ivorra_sof@gva.es

**Keywords:** fibromyalgia, pedometer, walking, self-report measures, physical activity

## Abstract

This study analyzes the degree of agreement between three self-report measures (Walking Behavior, WALK questionnaire and logbooks) assessing adherence to walking programs through reporting their components (minutes, rests, times a week, consecutive weeks) and their concordance with a standard self-report of physical activity (IPAQ-S questionnaire) and an objective, namely number of steps (pedometer), in 275 women with fibromyalgia. Regularized partial correlation networks were selected as the analytic framework. Three network models based on two different times of assessment, namely T1 and T2, including 6 weeks between both, were used. WALK and the logbook were connected with Walking Behavior and also with the IPAQ-S. The logbook was associated with the pedometers (Z-score > 1 in absolute value). When the behavior was assessed specifically and in a detailed manner, participants’ results for the different self-report measures were in agreement. Specific self-report methods provide detailed information that is consistent with validated self-report measures (IPAQ-S) and objective measures (pedometers). The self-report measures that assess the behavioral components of physical activity are useful when studying the implementation of walking as physical exercise.

## 1. Introduction

Fibromyalgia (FM) is a complex multidimensional disorder characterized by chronic diffuse musculoskeletal pain, low physical activity (PA) and a reduced quality of life [1,2,3]. An active lifestyle has been postulated as a resilience factor in FM [4,5], with exercise being one of the more commonly recommended non-pharmacological therapies for this population [2,3,4,5,6]. Recent studies have shown that increased exercise is a marker of health in FM [7,8,9] that improves the global well-being in this specific chronic pain problem [5,6]. In general terms, PA can include various tasks of daily living, such as work, mobility, leisure and recreational activities, that require musculoskeletal activity and energy expenditure. More specifically, exercise is a subset of PA and is defined as structured activity with a goal of improving physical performance and/or health [10].

In this line, walking is an effective way of exercise that is easy, accessible and with low musculoskeletal impact that has been widely described as a risk factor in cardiovascular disease [6] but is also related to physical function in chronic musculoskeletal pain. Focusing on chronic pain, walking has been included as one of the recommended primary treatments because of its low musculoskeletal impact [11,12,13] and its effect on pain relief, fatigue, anxiety, depression, number of falls, disability, balance, mobility and quality of life [8,9,11,12,13,14].

Unfortunately, FM patients rarely meet the guidelines for PA, including walking [15], and patients frequently avoid this behavior as a way of minimizing the pain they feel [16]. Patients argue that they feel an increase in pain when doing the activity, which leads them to stop [17]. Thus, considering that the FM population is mainly sedentary [18], it is especially recommended they start walking in a gradual way [19]. Specifically, for this population, the initial goal is to walk for a minimum of 30 min daily (in two bouts of 15 min each) and at least twice a week [10,11,12].

At the same time, pain is not the only factor that explains adherence to walking [19], and such disengagement makes normal daily life difficult, leading to loss of autonomy with respect to activities of daily living (ADLs) [19], and deprives people of opportunities to obtain positive reinforcement, which can lead to high physical disability and negative feelings [19,20,21]. Thus, besides pain, other symptoms such as fatigue, sleep problems, asthenia, cognitive alterations, gastrointestinal problems, anxiety, depression, migraine, paresthesia, imbalance and falls, among others, need to be considered [1,2,14,19,21] as part of the diagnostic criteria [3]. In fact, walking cannot be reduced solely to the individual’s capacity or to mere observable motor behavior. It is necessary to analyze this behavior in the context in which it takes place, taking into account the goals and meaning and purpose in life of patients in each context and determining variables in people’s quality of life and well-being [16,22,23,24].

In accordance with this, research focusing on factors associated with the discordance between how people with chronic pain feel about their abilities and how they actually perform [24] has called for increased attention to both the evaluation and implementation of multicomponent interventions that include objective and subjective measures, specifically those related to maintaining patients’ functioning despite ongoing pain [25].

Questionnaires that assess PA serve important purposes in both research and practical applications. These capture the self-reported adherence of the individual to the behavior, which is at times complemented with device-based measures [26]. In this line, researchers and healthcare professionals use steps as a clinically relevant objective for the classification of a “sedentary lifestyle” and for prescribing step-based PA recommendations in people with chronic pain [27,28]. Specifically, studies have included self-report measures as well as objective measures to assess PA, especially walking behavior [29,30,31]. Regarding subjective measures, self-reports are the most commonly used method because they are inexpensive and easy to administer. According to objective measures, assessment and interpretation of the number of steps taken per day (steps/day), measured by pedometers and accelerometers, have gained increasing acceptance by FM researchers [32] and have been recommended in international health guidelines for the FM population [33,34,35,36]. Moreover, pedometers have been included in many randomized controlled trials to analyze the effects of walking interventions in patients with chronic pain [19,37,38,39]. These electronic devices are becoming useful tools for recording and motivating behaviors related to PA [40], especially in interventions aimed at increasing PA, focusing on walking behavior [31,40,41,42].

The use of pedometers involves the setting of a step goal, the use of a device to register the steps and visual pedometer feedback [33,35,36]. In this sense, focusing on our FM population, pedometer-based interventions are highly recommended because their main purpose is to increase the number of daily steps—that is, walking. To achieve this increase in walking, researchers have defined two important elements: setting a step goal and the use of a step diary [33]. In this line, studies regarding interventions in chronic pain have shown that patients who are more sedentary before intervention programs are the ones who most benefit from them [33] and who most increase the number of steps walked in comparison to their baseline [43]. However, “benefits” and “steps walked” may not be precise representations of PA improvements, and when objective and subjective measures are compared, the results are not always in agreement [44]. Focusing on walking as a PA, researchers that have used a combination of objective and self-report measures of the environment have found them to be differently associated with walking levels [44].

In this line, the short version of the International Physical Activity Questionnaire (IPAQ-S) has been recommended to monitor patients because it integrates aspects of many areas of PA, allowing to record the frequency of activity in days per week and the duration in time per day as well as the values in real time for high-intensity activity, moderate-intensity activity and walking behavior. However, results from some studies have shown that when this questionnaire has been used to record walking behavior, it differs greatly in comparison with an objective measure [45].

Due to this, the assessment of walking in FM patients should not be limited solely to self-reports, and researchers must be mindful of clearly defining operationally how objective measures are assessing PA [46]. In this regard, researchers have pointed out that pedometers and accelerometers need to be included in multicomponent interventions as main measures related to walking because they provide an accurate and “objective” measure of number of steps [47] and provide feedback that becomes an important motivational tool to increase walking behavior [39,40,41,42,43,44,45,46,47,48]. However, these devices are mechanical and do not automatically rule out psychosocial factors that may affect the final performance [46]. Similarly, exercise and physical activity logbooks (or logs/diaries) have also been commonly used as a validation method for physical activity self-reports [49,50] and have shown medium-sized correlations with electronic measures and physical activity self-reports [51,52]. In an FM population, Zautra and Davis [53] found that logbooks of walking adherence measured for six weeks had significant and moderate correlations with the number of weeks the women walked for at least 30 min twice a week (r = 0.46, *p* < 0.000). According to this study, logbooks could be a valid indicator of walking, as another form of feedback, with more psychosocial factors included such as goals, emotions and cognitions to better understand the behavior [52,53].

Empirical evidence recommends that the goals established for each person have to be personalized according to their baseline reference values, their specific health aims and the sustainability of being able to carry it out as part of each person’s daily life [40]. In addition, it is necessary to achieve reliable assessment measures that help researchers to define adequate prevalence rates, to assess the needs of patients and, based on this, to be able to implement personalized treatments.

Thus, the aim of this article was to analyze the degree of agreement between three self-report measures (two questionnaires and logbooks) assessing adherence to walking programs through reporting their components (minutes, bouts, rests, times a week, consecutive weeks) and their concordance with a validated self-report measure of physical activity (IPAQ-S) and an objective measurement (pedometer).

## 2. Methods

### 2.1. Population

A total of 581 women from four Spanish FM associations were contacted and satisfied the inclusion criteria for this study: female, aged between 18 and 69 years, without a severe psychiatric comorbidity or other conditions preventing walking, meets the London-4 criteria for FM [54]. Participants were diagnosed either by rheumatologists or by primary physicians. However, as we did not have a second clinical diagnosis confirmation, the London-4 criteria were used to ensure population homogeneity because of its optimal sensitivity. Recruitment was performed via mail and phone. Out of the 581 eligible participants, we were unable to contact six and 122 refused to participate. Thus, our population comprised 453 women with FM, who were all contacted by ordinary mail, email and phone through the associations. Finally, 275 (47.2%) attended the appointment at their FM patients’ association or the university labs.

### 2.2. Measures

Walking behavior was measured by five different measures that are summed up in Table 1.

1. Self-reported adherence to a minimum walking program (Walking Behavior) [55] (measured at T1 and at T2). We asked the participants to indicate whether, in the past month and a half, they adhered to the components of the minimum walking program, namely ‘to walk with the aim of doing exercise, for at least 30 min, in bouts of 15 min with a small rest between bouts, at least twice a week over a minimum of six consecutive weeks’, as this is the recommended fixed program. We used two items rated on a seven-point scale (1–7), with the endpoints True/False and Definitively Yes/Definitively No [56]. The internal consistency scores were 0.93. We obtained the mean of the two items’ scores. Mean scores were computed, considering that higher scores indicated increased behavior. For this study, BehavT1 and BehavT2 represent the mean scores of items at T1 and seven weeks later (T2).

2. WALK Questionnaire (WALK) (measured at T1): A self-report of walking measured with four items asking about each component of the walking program (minutes, rests, times a week, consecutive weeks) during the last 6 weeks. As a result of combining those items, we calculated four binary variables in terms of whether the participant accomplished the following or not: the minimum walking program (walking for a minimum of 30 min in no more than two bouts of 15 min, at least twice a week, over a minimum of six consecutive weeks); just the minimum walking program; the standard recommended program (walking for over 50 min, in bouts of 15 to 20 min, with a small rest between bouts, four times a week, over a minimum of six consecutive weeks) and the maximum recommended program (walking for 60 min at least twice a week over a minimum of six consecutive weeks) [7] (see Table 1).

3. IPAQ-S questionnaire (IPAQ-S) (measured at T1): It is a measure of PA regarding the activity in the last week. The IPAQ short form (IPAQ-S) asks respondents to report the frequency and duration of walking, moderate-intensity activity and vigorous-intensity activity performed for at least 10 min per session. The IPAQ-S also collects information on total sitting time. Following the instructions, variables that measure days/week, minutes/day and minutes/week of each type of activity (intense activity, moderate activity, walking) and sedentary lifestyle were calculated [57]. Participants can also be classified into three levels of PA: “low” (physically inactive), “moderate” and “high” [58] (see Table 1).

The following measures were registered longitudinally over six weeks (from T1 to T2):

4. Logbooks with weekly sheets: Logbooks were created ad hoc for this study. Participants recorded in the logbooks the days of the week and dates. They were instructed to record the time at which they began to walk and the time at which they finished walking. Moreover, in each case, they indicated whether they had rested while walking (before 15 min, at 15 min, at 30 min or they have not had a break). For each day, they could complete two records of when they went for a walk at different times. They were given 6 logbooks to record information for 6 consecutive weeks. We transformed these raw data to obtain variables about walking during that 6-week period: number of times walking, number of weeks walking, number of weeks completing the minimum and the recommended program and two binary variables related to adherence to the minimum walking pattern during 4 or 6 weeks (see Table 1).

5. A pedometer (Yamax EX5103D USB pedometer) to measure walking behavior. The pedometer was worn only during walking for exercising. It collected information regarding average steps per day when walking for exercise. Yamax pedometers have a reputation throughout the world for accuracy and reliability [59,60]. Three-dimensional technology systems work in almost all positions. The three-dimensional sensor produces an electrical pulse in response to the movement of the body. The women could place the Yamax Power-Walker Pedometers in their pocket or purse and still obtain an accurate reading regardless of the position of the pedometer. The pedometers were given to the participants to register their walking behavior during six weeks (see Section 2.3). Participants were instructed to wear it consistently during walking hours. The raw data (steps a day) were transformed to calculate the average steps per walking day for exercising and the number of days that the patient walked 3000 steps or more.

### 2.3. Procedure

This study represents the first phase of a broader study aimed at increasing unsupervised walking in women with FM [55,61] (trial registration number: ISRCTN68584893). The study was approved by the Ethical Committee of the university. To select eligible participants, we sent letters to women from four fibromyalgia associations with a clinical diagnosis of FM (a requisite to join the association). We included information about the study, informed consent forms and questionnaires covering the variables related to the participation criteria. As we did not have a second clinical diagnosis confirmation, the London-4 criteria were used to ensure population homogeneity. In addition, we asked for information on physical comorbidities preventing walking and psychiatric diagnoses and treatments to explore the presence of major mental illnesses. A total of 581 members satisfied the inclusion criteria. Recruitment was performed via mail and phone. Out of the 581 eligible participants, we were unable to contact six and 122 refused to participate. Thus, our population comprised 453 women with FM who were all contacted by ordinary mail, email and phone through the associations. Finally, 275 (47.2%) attended the appointment at their FM patients’ association or the university labs (see Figure 1).

They all signed the informed consent form and filled out the self-reported adherence Walking Behavior, WALK and IPAQ-S questionnaires. We gave all participants logbooks to evaluate daily walking behavior during six weeks, and they received a new appointment 7 weeks later. We excluded the 20% of participants who did not attend the appointment or did not complete the logbooks from the final assessment (T2, *n* = 219). Furthermore, in order to obtain an objective measure of walking, we had 116 pedometers available to us. Finally, 109 patients agreed to use the pedometer. The pedometer protocol was only applied to the participants of two of the associations due to the number of pedometers available and the possibilities of applying the protocol. Instructions about the use of the pedometers and how to wear them were given along with written instructions for home. Participants wore the pedometer only when they walked for exercising over the period of six consecutive weeks. A researcher explained to each participant how to wear the pedometer specifically on the hand, giving them a demonstration. After the six weeks were completed (T2), the women were evaluated again using the self-reported adherence to Walking Behavior, and they also had to return the logbooks and the pedometers.

### 2.4. Statistical Analysis

In order to achieve the aim of this paper, the selection of the variables included in the three network models explained and presented below was based on the different variables and the two times of evaluation.

Two different times of evaluation were registered (T1 and T2), with six weeks between both, i.e., the period when women registered their walking behavior by both longitudinal measures: the pedometer and the logbook (see Table 1).

The first model included the self-reported adherence to Walking Behavior (T1), the WALK questionnaire and the IPAQ-S as a standard measure of habitual PA. The second model included the logbook, registering habitual walking activity, and the IPAQ-S. Finally, the third model included the two longitudinal measures to appraise walking behavior during the 6-week period between T1 and T2 (logbook and pedometers) and the self-reported adherence to Walking Behavior referring to those 6 weeks (T2).

Given the moderate presence of missing values, multiple imputation was performed using the mice package of the R environment [62]. The percentage of missing values was around 29.09% per variable (interquartile range = 48.45%). To approach relationships between variables, we implemented regularized partial correlation networks (RPCNs) as the analytic framework. Regularized partial correlation networks (RPCNs), also named as psychometric networks, are proposed as a technique to reliably estimate complex multivariate data [62]—more concretely, as a technique to explore associations between relatively large sets of variables without commonly known biases in standard correlation matrices, such as inflation of variance due to overlap between predictors. In addition, RPCNs are able to regularize false positive errors and estimate the robustness of results and the relative importance of each variable. This makes psychological data especially fit for RPCNs, since they commonly feature large sets of variables with relative overlap [63]. Therefore, RPCNs are suitable and promising for analyzing psychological variables. Moreover, RPCNs are very similar to structural equation modeling [50] but allow exploratory models to be made [64,65]. That is, they estimate correlations (named as ‘edges’) between variables (named as ‘nodes’) using correlation matrices as the source [64]. Then, they remove bias from each correlation with all other variables (partialization), in a similar fashion as multiple regression estimate slopes. In addition, RPCNs force small correlations to zero, assuming them as nuisance effects (regularization). Both partialization and regularization allow for a better control of false positives. The final result is a “curated” correlation matrix. This matrix is displayed graphically as a network of connected nodes, with more correlated variables in the center and less correlated variables in the periphery [63]. This process allows to explore relationships between variables in relatively large sets of variables. Among the existing RPCN estimation methods, the EBICglasso [66] method was selected due to its applicability to non-normal variables via non-paranormal transformation (NPNT) and to ordinal variables via polychoric or polyserial correlations [65]. EBICglasso uses the Extended Bayesian Inference Criteria (EBIC) to estimate correlations, along with the graphical least absolute shrinkage and selection operator (gLASSO) to regularize said correlations. Other models were discarded (e.g., mixed graphical model, Ising models) due to convergence issues or non-optimal adequacy. However, we also estimated the networks using regular correlations to assess the stability of results. Each network is displayed graphically with no restrictions to allocate the variables (or nodes). However, we also produced network plots forcing the same allocation of variables between methods for an easy comparison between standard correlations and EBICglasso methods. To interpret the estimated networks, the first step was a visual examination of the network to obtain the sign and magnitude of each correlation. However, the relative importance of each variable could be assessed with more accuracy using centrality indices. Centrality uses three main indices: ‘degree’ as the intensity of connections of each variable, measured as the sum of all non-zero edges for each node; ‘closeness’ as the amount of nodes with non-zero edges of certain nodes; and ‘betweenness’ as the degree of inter-connectedness of each variable, mediating between any pair of nodes. Centrality indices were estimated for each variable and ranked in standardized scores for easy interpretation. Finally, the stability and replicability of the network were assessed via non-parametric bootstrap of our estimated networks. This allowed us to estimate confidence intervals for each correlation (or edge) and each centrality index. All bootstraps were performed with 1000 samples. It is important to note that RPCNs aim to produce sparse models, preferring to produce false negatives than positives; thus, scarce correlations are to be expected. Sparsity was assessed with a sparsity index (i.e., the proportion of zero edges), with values around 0.5 as optimal 0.57. Three sets of RPCNs were produced: standard correlations, EBICglasso-NPNT maintaining the graphical set of the standard correlations and EBICglasso-NPNT allowing free graphical allocation. All analyses were computed using the network analysis module of JASP software (JASP team, Amsterdam, The Netherlands) [67], based on the bootnet package of the R environment [62].

## 3. Results

In terms of sociodemographic information, the mean age of our sample was 51.85 years (95% CI [50.75, 52.93]). In general, our sample was women with primary (47%) or secondary (28.10%) education that were working away from home (31%), housewives (26%) or unemployed (21.6%). Seventy-eight percent (*n* = 212) stated they had the medical recommendation to walk (see Table 2).

Focusing on networks, they were estimated and showed overall positive results (Figure 2), while the centrality estimates showed apparently discriminative results (Table 3).

The first network (Figure 2, Section a) showed a positively correlated network but with some negative edges between variables. As expected, same-instrument correlations were stronger than different-instrument ones, although they were mostly negative among the WALK questionnaire. Shifting to the RPCN increased sparsity (from 0.000 to 0.859). The strongest same-instrument connections were within the IPAQ-S, with the highest value between TotMinWeek and Wdays (r = 0.37) and Mdays (r = 0.31). The strongest correlations for the WALK questionnaire were between RECw and justMINw (r = −0.39), followed by MINw with RECw (r = −0.17). Different-instrument connections remained positive. More concretely, WALK and Walking Behavior were connected, and WALK was also related to IPAQ-S.

Regarding centrality (see Table 3), TotMinWeek seemed to be the most central, with high betweenness and degree indices, followed by MINw and Wdays for betweenness. In addition, SIT, Mminutes and Vminutes showed low degree indices but intermediate scores in betweenness.

The second network (Figure 2, Section b) showed high positive connections. Again, same-instrument connections were stronger than the different-instrument ones. Shifting to RPCN increased sparsity (from 0 to 0.699). The highest same-instrument connections were within the IPAQ-S, showing the highest connection between AFCat and TotMinWeek (r = 0.53), followed by TotMinWeek with Wdays (r = 0.27) and Mdays (r = 0.21). Logbooks showed strong connections between TF30P and NWeekP (r = 0.45) and between NWeekP and P90 (r = 0.37). Different-instrument correlations were scarce, with only two relevant positive connections: TFrec with Wdays (r = 0.05), and TTime with Wminutes (r = 0.06).

Centrality results were mixed (see Table 3). TFrec showed high betweenness and closeness, followed by Wdays, but TTime also showed high betweenness and degree, followed by TF30P, although TF30P showed the highest overall degree. On the other hand, Mminutes and SIT showed the lowest scores on closeness and degree, followed by Vminutes showing the lowest overall degree. Thus, the most central nodes were also connected between instruments.

The third network (Figure 2, Section c) was also highly connected and positive and increased its sparsity when shifting to RPCN (from 0.000 to 0.244). Same-instrument variables maintained greater connections than different-instrument ones. The highest same-instrument connections were for the pedometer (NDaysWalk and NDaysWalk3000, r = 0.58), with logbooks following in intensity (i.e., TF30P and NWeekP, r = 0.51; TF30P and TFrec, r = 0.40). Positive different-instrument connections were shown between logbook measures and pedometer (TFrec with NDaysWalk: r = 0.10) and Walking Behavior (NWeekP with BehavT2: r = 0.16), but not between pedometer and Walking Behavior.

Regarding centrality (see Table 3), TF30P and NWeekP seemed to be the most central in all indices. In addition, NDaysWalk and NDaysWalk3000 showed especially low closeness, which added to the high correlation between them, providing evidence of being isolated variables. Finally, Walking Behavior showed intermediate levels of betweenness and closeness but showed the lowest degree of all, thus being weak and isolated.

As expected, same-instrument connections (i.e., correlations or weights) remained stronger than different-instrument ones.

## 4. Discussion

The aim of this article was to analyze the agreement between three subjective measures used to assess walking (self-reported adherence to Walking Behavior, WALK questionnaire and logbooks) as well as the concordance between these three subjective measures of walking with a validated self-report measure (IPAQ-S) and an objective measurement (pedometer). In order to achieve this aim, three different models were created. The WALK and IPAQ-S questionnaires were applied at the first time point (T1), while pedometers (steps) and logbooks were applied from T1 to T2 (6 weeks). Self-reported adherence to Walking Behavior was evaluated in both periods, T1 and T2, and they were included in the analysis according to the aim of the study. These three models were highly interconnected and showed positive networks. Our first model included variables from three self-report instruments; two measures of walking, namely the self-reported adherence to Walking Behavior (BehavT1) and WALK; and a standard measure of PA which includes walking (IPAQ-S). The results have shown firstly that both self-report measures that evaluate the adherence to the components of the minimum walking program are connected (BehavT1, MINw). Secondly, self-reports of adherence to this program are connected with the walking activity from the IPAQ-S (Wdays). Both measures are coherent, as when patients reported that they adhered to a minimum walking program, they also reported compliance with the minimum components required in this program. In addition, the short version of the IPAQ supports the validity of this self-report according to the adherence to a minimum walking pattern. As has been pointed out, the short-form IPAQ has the advantage of assessing compliance with PA guidelines [42]. However, it shows disadvantages, such as the difficulty to recall PA details [68,69]. Thus, the self-reporting of components of walking (minutes, bouts, rests, times a week, consecutive weeks) is a good measure of which specific walking program patients maintain. The validity of this measure is supported by its connection with the IPAQ-S, which can complement the evaluation of study outcomes.

With the aim of analyzing the self-reported habitual walking activity, the second model integrated the logbooks and the IPAQ-S. This model showed a highly interconnected and positive network, where both measures are connected. They are independent measures, both useful to measure walking specifically. In particular, the total number of minutes and times the patients walked as reported in the logbooks (TTime, TFrec) are connected with days walking a week and minutes walking a day as evaluated by the IPAQ-S (Wdays, Wminutes). Moreover, regarding centrality, these variables are the most important within the network itself. Taking into account that RPCNs are prone to false negatives, they can be used as a selection process or for screening of relationships. Therefore, those walking variables that remain can be proposed as candidates to be selected for a more summarized use of the instrument.

We underline the coherence of these results and, again, the usefulness of assessing walking behavior in detail. The use of logbooks as a form of daily reporting of walking was supported by the IPAQ-S, specifically related to walking. Furthermore, moderate and vigorous physical activities from the IPAQ-S did not show connections with the logbook measures. In fact, the isolated variables in the model were those related to sitting and to moderate or vigorous physical activities. This could be expected considering that logbooks report walking but not other physical activities. The results were also in agreement with studies that had pointed out that the IPAQ-S reports acceptable test–retest reliability for walking in FM populations [51] but not for moderate- and vigorous-intensity activities [45]. The fact that the isolated variables in this model are those related to sitting and moderate- and vigorous-intensity activities shows that logbooks and the IPAQ-S together are useful to evaluate specifically what we were focusing on: a walking program in FM patients.

Finally, the third model included the two longitudinal measures to appraise walking behavior during six weeks (logbook and pedometer) and the self-reported adherence to Walking Behavior during six weeks. The results showed a strong connection between the number of weeks that patients achieved the minimum walking program (NWeekP, logbook) and their reports of the self-reported adherence to Walking Behavior (BehavT2). As in previous results, the self-reported adherence to Walking Behavior is a measure capable of providing a valid estimate of walking [61], showing agreement in this study with both daily records and self-reported adherence to components of the walking program. As in the first model, the results support the coherence of the women’s perception of their adherence to a detailed minimum walking program with, in this case, the specification of daily walking activity.

Walking in accordance with the minimum program (NWeekP, logbook) also showed a strong connection with the frequency of walking (TFrec, logbook), which were the most central variables of the model evaluated through the logbook. In addition, this frequency of walking was positively connected to the frequency of days walking assessed with the pedometer. Although both pedometer variables were quite isolated in the network, our results showed that in contrast to studies that provided different estimates between self-reported and objective measures [69], both longitudinal measures, the pedometer and the logbook, are capable of providing a valid estimation of frequency of walking [48]. Thus, the self-reported adherence to Walking Behavior is able to provide a valid estimate of the recommended minimum program in FM patients, the logbook is able to provide a valid estimate of walking and the pedometer, as a form of objective assessment in this study, supports the validity of daily self-reports of walking using logbooks.

Focusing on a clinical setting, studies have suggested that walking improves pain [70,71], physical function [70,72], overall well-being [73] and symptoms [74,75,76] in FM. Thus, we argue the need to complement measures of walking included in this study. Interventions with the aim of promoting walking should assess the individual contexts in which the behavior takes place. Those aspects may be useful to identify clinical intervention profiles of women with FM to improve the efficacy of walking interventions [19].

### Limitations

Firstly, we included pedometers in this study as an objective measure of study, and they were used in a sub-sample. Even though the limitations of the first pedometers have been resolved by technological advances in recent years, this was done because pedometers are a notably efficient tool to achieve an increase in PA [36,37,38,39,77]. Furthermore, the size of the sub-sample was large enough to reach the goal of comparing it with the logbooks. In fact, our results show that both objective and self-report measures are connected. Complementary to the objectives of the present study, future studies could analyze the concordance between self-reported measures and other technologies such as accelerometers.

Secondly, in the second model a longitudinal measure, the logbook, was analyzed in comparison to IPAQ-S, a measure taken at the initial time point. However, the IPAQ-S refers to habitual behavior exercise, which is why it was evaluated only at T1. Our results show that both measures are connected in terms of habitual walking program.

Thirdly, we used convenience sampling, which results in some limitations such as lower representativeness of the Spanish population with FM and unknown levels of sampling error. Otherwise, our sample population was between 18 to 69 years old and fulfilled the London-4 criteria [54,78], which also included ages for which the IPAQ-S was designed, adding to the validity of our results. However, the absence of a male sample did not allow us to know whether these findings apply to men, so future studies should analyze the validity and reliability of the measures used in men with FM. Another point to highlight is that based on the inclusion criteria, although the age limit for participation in the study was 69 years, the mean age was high, and the educational level was mostly primary. Nevertheless, in order to prevent possible comprehension problems regarding the logbooks and the use of the pedometer, it was verified that participants understood the instructions, and the evaluation was carried out with the support of professionals trained in the administration and use of the measures used.

We have to point out that asking the participants to indicate whether, in the past month and a half, they adhered to the components of the minimum walking program led to identifying the key components to be targeted in interventions that promote walking as physical exercise [79]. As a strong point, we included not only this specific self-reported adherence measure of walking but also the specification of how a person performs each component (WALK) and the daily register of walking (logbook). Our results support the use of these self-reported measures as their specifications provide useful instruments to use in motivational theory-based interventions aimed at promoting changes in walking behavior [79].

## 5. Conclusions

This report is another step in developing best practices for using measures of PA assessment in FM. First, we can conclude that when behavior is assessed specifically and in detail, the results of the different self-report measures are in agreement. Self-report measures that assess detailed walking programs, specifying their behavioral components (Walking Behavior, WALK), and daily diaries (logbook) are useful and reliable as an outcome measure in studies of implementation of walking as physical exercise.

Second, self-report methods (Walking Behavior, WALK, logbook) provide detailed information that is consistent with validated self-report measures (IPAQ-S) and objective measures (pedometers). The women with FM showed agreement between their reports of adherence to a minimum walking program and the minimum components required in this program.

Third, to prevent the consequences of a sedentary lifestyle, increasing daily PA levels according to a recommended program is one of the main aims of interventions in FM. Studies have indicated that FM has more of an impact on the self-perceived ability to perform physical activity than on performance of the activity itself [26,80,81,82]. However, measures reflecting the prescribed behavior might eliminate an overestimation of the PA carried out [83]. Thus, it is necessary to include validated measures to establish a baseline of walking behavior, as presented here (Walking Behavior) [61,84].

Finally, based on recent contextual models of chronic pain [85], FM impact may be influenced by other contextual psychological factors that could have a moderating influence on patient perception and symptoms [86]. Including different self-report measures, complemented by logbooks, to assess walking may be beneficial to capture contextual factors related to the walking program and, specifically, associated with the minimum walking program in FM samples, along with assessments by objective measures.

## Figures and Tables

**Figure 1 ijerph-19-02995-f001:**
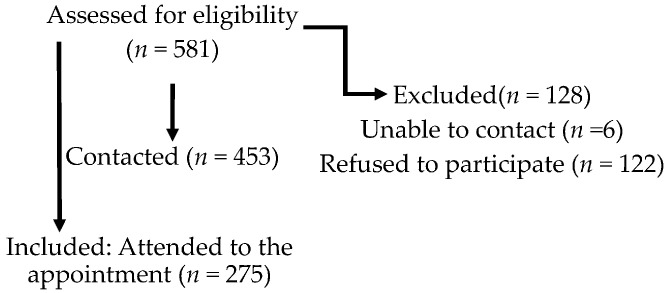
Flow diagram of the study participants.

**Figure 2 ijerph-19-02995-f002:**
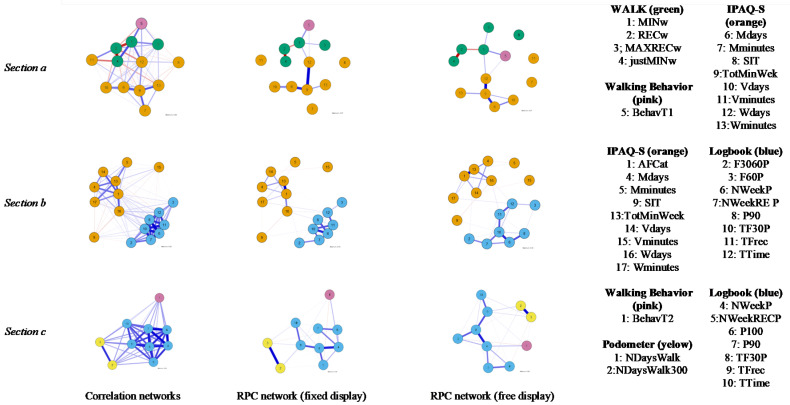
Regularized partial correlation networks for the WALK, Walking Behavior, IPAQ-S, logbook and pedometer measures.

**Table 1 ijerph-19-02995-t001:** Walking measures and variables.

**Time 1 (T1)**
**Walking Behavior** BehavT1: Mean score of self-reported adherence to minimum walking program
**WALK questionnaire (WALK)** MINw: Complete walking minimum program: 0 ‘NO’, 1 ‘YES’ justMINw: 0 ‘Complete more than minimum program’, 1 ‘Just the minimum program’ RECw: Complete walking recommended program: 0 ‘NO’, 1 ‘YES’ MAXRECw: Complete maximum recommended program: 0 ‘NO’, 1 ‘YES’
**IPAQ-S Questionnaire (IPAQ-S)** Wdays: ‘Walk (days/week)’ Wminutes: ‘Walk (minutes/day)’ Mdays: ‘Moderate-intensity activity (days/week)’ Mminutes: ‘Moderate-intensity activity (minutes/day)’ Vdays: ’Vigorous-intensity activities (days/week)’ Vminutes: ‘Vigorous-intensity activities (minutes/day)’ TotMinWeek: ‘Total activity (minutes/week)’ SIT: ‘Sit (minutes/week)’ AFCat: Categorical variable; low, moderate and high level of physical activity
**Time 2 (T2)**
**Walking Behavior** BehavT2: Mean score of self-reported adherence to minimum walking program
T1–T2
**Logbook** TFrec: Total times walking TTime: Total minutes walked F3060P: Number of times/week between 30–60 min F60P: Number of times/week > 60 min TF30P: Total times walking minimum program (30 min) NWeekP: Number of weeks minimum program completed NWeekRECP: Number of weeks of walking recommended program P90: Completed 90% of walking minimum program (4 weeks) P100: Completed 100% of walking minimum program (6 weeks)
**Data of Pedometers: Steps** NDaysWalk: Average steps per walking day for exercising NDaysWalk3000: Number of days that patient walked 3000 steps or more

**Table 2 ijerph-19-02995-t002:** Participant demographics data.

**Age**	**Medical Status**
Mean age: 51.85 (95% CI [50.75, 52.93])	78% had the medical recommendation
SD = 9.16, Mdn = 52.69	to walk (*n* = 212)
**Employment status**	**Education**
31% working away from home (*n* = 85)	12% university education (*n* = 33)
26% housewives (*n* = 71)	12.8% literate (*n* = 35)
21.6% unemployed (*n* = 59)	47% primary education (*n* = 129)
9.9% retired due to pain (*n* = 27)	28% secondary education (*n* = 77)
6.6% on sick leave (*n* = 18)	
4.8% retired (*n* = 13)	

**Table 3 ijerph-19-02995-t003:** Standardized centrality scores.

		Betweenness	Centrality Closeness	Degree
(a)	**Walking Behavior**			
	BehavT1	−0.667	0.000	−0.377
	**WALK questionnaire**			
	MINw	2.00 *	0.000	0.768
	RECw	0.189	0.000	0.768
	MAXRECw	−0.667	0.000	−0.77
	justMINw	−0.667	0.000	0.162
	**IPAQ−S**			
	Mdays	0.189	0.000	0.621
	Mminutes	−0.667	0.000	−1.13 *
	SIT	−0.667	0.000	−1.13 *
	TotMinWeek	1.47 *	0.000	2.199 *
	Vdays	−0.667	0.000	−0.107
	Vminutes	−0.667	0.000	−1.13 *
	Wdays	1.47 *	0.000	0.766
	Wminutes	−0.667	0.000	−0.637
(b)	**Logbook**			
	NWeekP	−0.137	0.367	1.08 *
	NWeekRECP	−0.396	0.191	0.660
	F3060P	−0.862	−0.061	−0.696
	F60P	−0.862	−0.613	−1.00 *
	TF30P	1.41 *	0.765	1.82 *
	TFrec	1.88 *	1.00 *	0.640
	TTime	1.62 *	0.970	1.31 *
	**IPAQ−S**			
	AFCat	0.640	0.743	1.23 *
	Mdays	−0.758	0.211	−0.075
	Mminutes	−0.862	−2.04 *	−1.26 *
	P90	−0.862	−0.129	−0.161
	SIT	−0.862	−1.00 *	−1.17 *
	TotMinWeek	0.174	0.562	0.356
	Vdays	−0.085	−0.053	−0.467
	Vminutes	−0.862	−2.35	−1.40 *
	Wdays	1.364 *	1.05 *	−0.098
	Wminutes	−0.551	0.392	−0.763
(c)	**Walking Behavior**			
	BehavT2	−0.702	−0.890	−1.54 *
	**Pedometer**			
	NDaysWalk	0.000	−1.09 *	−0.177
	NDaysWalk3000	−0.702	−1.34 *	−0.062
	**Logbook**			
	NWeekP	1.30 *	1.18 *	1.50 *
	NWeekRECP	−0.702	0.158	−0.321
	P100	−0.702	−0.554	−0.611
	P90	−0.702	0.013	−0.279
	TF30P	1.70 *	1.49 *	1.74 *
	TFrec	1.20 *	1.10 *	0.488
	TTime	−0.702	−0.083	−0.733

Standardized centrality scores (z-scores). * Z-score > 1 in absolute value. (a), (b) and (c) reflect networks 1, 2 and 3, respectively.

## Data Availability

Data are available by request to the corresponding author (carmen.ecija@urjc.es).

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
