# Peer review of "Assessing Walking Programs in Fibromyalgia: A Concordance Study between Measures"

_ijerph, 2022, doi:10.3390/ijerph19052995_

Round 1
Reviewer 1 Report
ijerph-1584750_review
Title: Assessing walking programs in fibromyalgia: a concordance study between measures.
Comments and Suggestions for Authors
Dear authors,
I have carefully read your paper, which analyse the degree of agreement between three self-report measures assessing the adherence to walking programs through reporting its components, its concordance to a validated self-report measure of physical activity, and to an objective measurement (pedometer) in women with Fibromyalgia.
In my opinion, you have responded positively to the suggestions for improvement made, you have expanded the information required in the introduction and material and methods sections; substantially improved the results and has also added a flow diagram to facilitate monitoring the entire study participants.
I would like to comment only a minor issue that could be addressed to improve the document, in my opinion.
Specific comments:
Material and methods
- Page 5, lines 197-198. In the original version of your manuscript you mentioned that “and represents the first phase of a larger study (trial registration number: ISRCTN68584893). The study was approved by the Ethical Committee of the Miguel Hernández University” this information is relevant, I recommend that you add this information in the revised version.
I believe that all these modifications have improved the quality of your manuscript.
Therefore, I congratulate you on your great effort and the work you have done.
Author Response
February 24th, 2022
Dear Reviewer,
On behalf of our research team, I would like to thank you for all these comments about our study (reference: IJERPH-1584750), and we really appreciate your specific suggestions.
We have introduced the trial registration number (line 214), and the reference of the ethical committee of the university (line 522).
Yours sincerely,
Carmen Écija, PhD (corresponding author)
Department of Psychology.
Rey Juan Carlos University.
28922 Madrid (Spain)
Phone: 0034-914888943/0034679164381
Carmen.ecija@urjc.es

Reviewer 2 Report
Dear Editor,
the difficulty in quantify the disability of patients affected by fibromyalgia in walking is an "open topic", no guidelines nor protocols are present in the current literature.
I recommend
- shortening the introduction to make it more fluid.
- adding the role of autonomy in ADL in the conduction of the study
- deepening the imbalance in fibromyalgia
- Chiaramonte R, Bonfiglio M, Chisari S. Multidisciplinary protocol for the management of fibromyalgia associated with imbalance. Our experience and literature review. Rev Assoc Med Bras (1992). 2019 Nov 7;65(10):1265-1274. doi: 10.1590/1806-9282.65.10.1265.
- Radunović G, Veličković Z, Rašić M, Janjić S, Marković V, Radovanović S. Assessment of gait in patients with fibromyalgia during motor and cognitive dual task walking: a cross-sectional study. Adv Rheumatol. 2021 Aug 26;61(1):53. doi: 10.1186/s42358-021-00212-5
Author Response
February 24th, 2022
Dear Reviewer,
On behalf of our research team, I would like to thank you for all the comments about our study and the opportunity to improve the manuscript (reference: IJERPH-1584750). Our point-by-point response is outlined below.
Considering these comments, we have checked the introduction trying to reduce contents as well as changes suggested from the other reviewers to better understand differences between types of PA.
We have introduced both references, incorporating imbalance and falls as important variables to take in account in this impaired illness. Moreover, trying to introduce the role of autonomy in activities of daily living (ADLs) as you have recommended, we have highlight how pain and symptoms derived from the illness could influence ADLs from women with FM. Loss of autonomy implies not only negative feelings, but it also has an important influence on their quality of life and well-being. We think this comment help us to introduce benefices of walking not only on pain. Walking may also improve patients’ autonomy, quality of life and well-being. This information is included in lines 56, 57 and 62-65. References 16 and 21.
We really appreciate your suggestions and hope that these manuscript improvements respond to them.
Yours sincerely,
Carmen Écija, PhD (corresponding author)
Department of Psychology.
Rey Juan Carlos University.
28922 Madrid (Spain)
Phone: 0034-914888943/0034679164381
Carmen.ecija@urjc.es

Reviewer 3 Report
Abstract Section the method objective Present p values when the analysis indicates significance Introduction Line 25 refers to recent studies, but the studies cited date from 2017, change the text or add studies less than 3 years old. Since the theme is based on the benefits of walking, explore in a clear, summarized and precise way the physiological, endocrine-humoral and psychological benefits of this type of physical activity. Methods How was the clinical analysis of severe mental illness excluded? Who applied the questionnaires received training to do so? Explain how to calculate questionnaire scores In the case of Logbooks filled in poorly or with incomplete information, how did you proceed? In the table, remove the signs (circles) from the lines, as the reading was impaired, and change the name from table to table, as that is what it is about. On line 198, present the ethics committee approval number How long did each volunteer take the entire analysis? Results In table 2, remove the signs (circles) from the lines, as the reading was impaired, put the information in 2 columns In table 3, remove the signs (circles) from the lines, as the reading was impaired, discussion Discuss the use of Logbooks in underserved populations in culture and education Discuss the use of Logbooks or pedometer for elderly populations over 70, whose digital and memory difficulties can harm, suggest options?
Author Response
February 24th, 2022
Dear Reviewer,
On behalf of our research team, I would like to thank you for all the comments about our study and the opportunity to send a reviewed version of the manuscript (reference: IJERPH-1584750). We have considered all of your comments and we think the manuscript has been improved. Our point-by-point response is outlined below.
REVIEWER 3
English language and style are fine/minor spell check required
We have checked the language
Abstract. Section the method objective.
- Present p values when the analysis indicates significance
According with your comment about significance, asterisks in table 3 indicate that the z score is greater than 1 standard deviation (i.e., > 1). We did not use significance in the analyses. As you have suggested, we have also included this information in the abstract (line 22).
Introduction
- Line 25 refers to recent studies, but the studies cited date from 2017, change the text or add studies less than 3 years old.
Thank for your suggestion. We have proceeded to include new actualized references from three studies from 2020 and 2021. Specifically, we have added new references 7, 8 and 9 (line 36).
- Loftus, N.; Dobbin, N.; & Crampton, J. S. The effects of a group exercise and education programme on symptoms and physical fitness in patients with fibromyalgia: a prospective observational cohort study. Rehabil., 2021, 0, 1–8. https://doi.org/10.1080/09638288.2021.1891463
- Bidonde, J.; Boden, C.; Foulds, H.; Kim, S.Y. Physical Activity and Exercise Training for Adults with Fibromyalgia. In Fibromyalgia Syndrome; Ablin, J.N., Shoenfeld, Y., Eds.; Springer: Cham, Switzerland, 2021; pp. 59–72.
- Andrade, A.; Dominski, F.H.; Sieczkowska, S.M. What we already know about the effects of exercise in patients with fibromyalgia: An umbrella review. Arthritis Rheum. 2020, 50, 1465–1480.
- Since the theme is based on the benefits of walking, explore in a clear, summarized and precise way the physiological, endocrine-humoral and psychological benefits of this type of physical activity..
Following reviewer comment, new information of walking related to physiological and psychological benefits of this type of physical activity has been added (lines 41 to 47). Firstly, widely information about cardiovascular disease have been included, following of information related to walking in chronic pain population. As reviewer comment, we had not explained specifics benefices from walking. We agree with the comment mentioned; information added give the introduction more coherence with our aim. However, information about endocrine-humoral have been not included because this specific information is not our scope and not essential for the main aim of the study. References included are 8, 9, 14 and 15. Moreover, according to comments from reviewer 2, new information about how walking affects autonomy, and how management of fibromyalgia is associated with imbalance of patients have been also included. This information is focused on how walking improve quality of live and well-being from chronic pain patients, reducing also number of falls, and improving imbalance. This information is related to psychological and physiological benefits of this type of physical activity. References from this information have been included (16 and 21) after information described (lines 56-57 and 61-66). We have tried to summarized information include, trying to include the important information required by every reviewer.
Methods
- How was the clinical analysis of severe mental illness excluded?
To select the eligible participants, we previously had sent letters to women of the four Spanish fibromyalgia associations with: information about the study, informed consent forms, the London-4 criteria for FM and other questionnaires covering the remaining variables related to the participation criteria. Specifically, we asked for psychiatric diagnosis and treatment in order to explore the presence of major mental illness. We have completed this information in 2.3. Procedure (lines 216-222).
- Who applied the questionnaires received training to do so?
Evaluators were psychology students well trained people to apply questionnaires
- Explain how to calculate questionnaire scores
We have modified the explanation in 2.2. Measures in order to clarify this issue.:
In measure 1 we obtained the mean of the two items scores (lines 164-165)
In measure 2 we have clarified the explanation (lines 170-178)
Measure 3: IPAQ-S variables were calculated following instructions of authors, that is, cleaning data procedure, mistakes responses elimination, and data transformations. Better reference from this instruction has been included [58].
Measure 4: Raw data of logbooks were introduced in data base. And data transformations let to obtain for each participant variables about walking during the six weeks, which are shown in Table 1. Two of these were binary variables related to accomplishment or no the walking minimum program. We also have modified the explanation (lines 195-199).
Measure 5: The raw data of pedometers (steps a day) were transformed to calculate the average steps per walking day for exercising and the number of days that patient walked 3000 steps or more (lines 209-211).
- In the case of Logbooks filled in poorly or with incomplete information, how did you proceed?
We have excluded in the study the participants who did not complete the information of Logbooks. In T1 we had n= 275; 56 women did not attend the study in T2 and/or did not complete the information (lines 235-236).
- On line 198, present the ethics committee approval number
Following the journal's template, this information has been included in the Institutional Review Board Statement at the end of the document (lines 522-523).
Results
- In the table 1, remove the signs (circles) from the lines, as the reading was impaired, and change the name from table to table, as that is what it is about.
- Results In table 2, remove the signs (circles) from the lines, as the reading was impaired, put the information in 2 columns
- In table 3, remove the signs (circles) from the lines, as the reading was impaired
Following review comment, circles has been removed from both tables. After careful review of the template provided by the journal, the tables have been adjusted to the established format.
Discussion
- Discuss the use of Logbooks in underserved populations in culture and education
- Discuss the use of Logbooks or pedometer for elderly populations over 70, whose digital and memory difficulties can harm, suggest options?
Information required from both comments has been included in the limitations of the study. Specifically, we have highlight that, based on the inclusion criteria, although the age limit for participation in the study was 69 years, the mean age is high, and the educational level is mostly primary. Nevertheless, to prevent possible comprehension problems regarding the logbook and the use of the pedometer, we have explained that it was verified that participants understood the instructions, and the evaluation was carried out together with support professionals trained in the administration and use of the measures used. (Lines 471-477).
We really appreciate all the comments and hope that these manuscript improvements respond to them.
Yours sincerely,
Carmen Écija, PhD (corresponding author)
Department of Psychology.
Rey Juan Carlos University.
28922 Madrid (Spain)
Phone: 0034-914888943/0034679164381
Carmen.ecija@urjc.es

Round 2
Reviewer 2 Report
I appreciated the effort of improving the manuscript. It is a very interesting and well-written paper.